# Temporomandibular Disorders in Patients with Rheumatoid Arthritis

**DOI:** 10.3390/jcm14207381

**Published:** 2025-10-18

**Authors:** Anna Wydra-Karbarz, Zbigniew Guzera, Bogdan Batko, Mateusz Moskal, Katarzyna Błochowiak

**Affiliations:** 1Dental Office Relax Dental Spa, 26-600 Radom, Poland; 2Świętokrzyskie Rheumatology Center, Health Care Center, 26-200 Końskie, Poland; 3Department of Rheumatology and Immunology, J.J. Specialist Hospital Dietla, 31-121 Krakow, Poland; 4Department of Rheumatology and Immunology, Faculty of Medicine and Health Sciences, Andrzej Frycz Modrzewski Krakow University, 30-705 Kraków, Poland; 5Department of Oral Surgery, Poznan University of Medical Sciences, 60-812 Poznan, Poland

**Keywords:** temporomandibular joint, temporomandibular disorders, rheumatoid arthritis, myofascial pain, orofacial pain, joint crepitation, jaws, disease activity score

## Abstract

**Background and Objective:** Rheumatoid arthritis (RA) is a systemic autoimmune disease affecting synovial joints including the temporomandibular joint (TMJ). This study aimed to assess the prevalence and characteristics of temporomandibular disorders (TMDs) in RA patients and correlate these findings with disease activity markers. **Materials and Methods:** This cross-sectional study included 40 RA patients meeting the 2010 ACR/EULAR criteria and 40 healthy subjects (HSs). Research diagnostic criteria for TMD were used to assess TMD. RA severity was evaluated using ESR, CRP, rheumatoid factor (RF), anti-CCP antibodies, Disease Activity Score (DAS) 28, and disease duration. **Results:** TMD prevalence was significantly higher in RA patients (75%) than in HS. Orofacial pain was a predominant TMD reported in 82.5% of RA patients. In RA patients there was a difference in myofascial pain, TMJ pain, and TMJ sounds in comparison to HS. All masticatory muscles were painful on palpation in RA patients and the pain intensity was higher in RA than in HS. The most painful muscles were the medial pterygoid muscles, the anterior belly of digastric muscle, and the tendon of the temporal muscle. Slight crepitations were the most frequent. Maximal active mouth opening was reduced and negatively correlated with CRP levels. The most frequent jaw function limitations were chewing and yawning difficulties and tinnitus. There were no correlations between TMD and DAS, RF, and disease duration. **Conclusions:** Active inflammation in RA is a crucial factor reducing mouth opening. TMD screening independent of disease duration should be integrated into RA management protocols, particularly for patients with elevated inflammatory markers, to eliminate other pathological factors contributing to faster TMJ functional changes, TMJ involvement, and the severity of TMD during RA course.

## 1. Introduction

Rheumatoid arthritis (RA) is a systemic autoimmune disease characterized by chronic inflammation and symmetric involvement of synovial joints. It is one the most common rheumatic disease, affecting 0.5–2% of the population worldwide [1]. Its etiology is multifactorial and complex, comprising genetic predisposition, immune systems disturbances, and environmental factors. The most common clinical findings include joint pain and swelling, morning stiffness, malaise, and fatigue [1,2,3]. Moreover, RA induces some changes in blood count test, production of pro-inflammatory cytokines, and specific serum antibodies particularly for rheumatoid factor (RF) and citrullinated peptide (anti-CCP), and increase in laboratory parameters reflecting inflammatory status, including C-reactive protein (CRP) and erythrocyte sedimentation rate (ESR) [1,4]. Furthermore, the values of ANAs (antinuclear antibodies), Hb (hemoglobin), PLT (platelets), and WBC (white blood cells) may also be sought in RA [1]. However, approximately 20% of RA are seronegative [1]. Although RA may show spontaneous remission, long-standing disease symptoms and persistent inflammation lead to progressive and irreversible damage to numerous synovial joints and disability. The histopathological basis of RA is hyperplasia of the synovial lining, leading to pannus formation and subsequent degradation of the cartilage matrix and the subchondral bone, promoting joint destruction [1].

The temporomandibular joint (TMJ) is a synovial joint that is prone to RA. However, the frequency and degree of TMJ involvement in RA ranges widely from 4 to 86%, depending on the study group, diagnostic criteria, and methodology used in the study [1]. It is estimated that more than 50% of the RA patients present clinical symptoms of TMJ involvement [1]. However, TMJ involvement in RA is underestimated because the temporomandibular joint rarely exhibits classical signs of inflammation, in comparison to other synovial joints, and manifests as subjective symptoms not correlating with real imaging and morphological findings. The most reliable method to diagnose active TMJ arthritis in RA is enhanced magnetic resonance imaging (MRI). Unfortunately, this way of diagnosing is costly, time-consuming, and not readily available everywhere. Therefore, it is not routinely performed in RA patients for the diagnosing of active TMJ arthritis. Moreover, there is no consensus as to the relationship between TMJ involvement and inflammatory parameters, severity of RA, and its duration. There are many local as well as systemic potential factors that might modulate and modify severity of TMJ disorders in RA. Moreover, the uniqueness of the TMJ and its close functional connection with the masticatory muscles and other components of the stomatognathic system might cause pathologies of the joint induced by RA to manifest in a different way in comparison to other synovial joints. This makes TMJ involvement in RA more challenging.

All these manifestations are collectively defined as temporomandibular disorders (TMDs). They represent a heterogenous group of inflammatory or degenerative symptoms of the stomatognathic system and dysfunctional clinical features, involving TMJ and related masticatory muscles. They may affect as many as 5–12% of the whole population and are the second most frequent complaints reported by patients, immediately after back pain [5]. The most common clinical findings of TMD are pain in the TMJ area, facial pain, headache, tenderness of the masticatory muscles, asymmetry in the jaw movement, joint sounds such as crepitation, limitation in mouth opening and jaw function, masticatory difficulties, and altered occlusion. Less frequent disorders comprise earache, tinnitus, vertigo, and neck, shoulder, and back pain. Most of the observed symptoms are typical of other musculoskeletal and rheumatologic disorders or are results of disorders affecting the TMJ itself [5,6]. There are no data to verify if RA is predisposed to a specific type of TMD and how high the risk of TMD is in RA. Deep analysis of the type of TMD in RA may slow down the progression of degenerative changes in TMJ. Identification of RA-specific TMD may facilitate the development of guidelines and recommendations for the management of RA patients in order to counteract the most adverse symptoms of the disease and to increase awareness of TMD in RA patients. One of the most severe manifestations of TMJ involvement in RA, described in previous studies, is malocclusion of the teeth and anterior open bite, which usually occur in advanced stages of RA. Some of the morphologic changes typical of RA can be detected on conventional radiographs, cone beam-computed tomography (CBCT), and MRI of TMJ. They include spiked deformity or pencil-like condylar processes, cortical erosion, narrowing of the joint disc and its perforation, and also a tendency to more frequent erosive changes, subchondral sclerotization, and flattening of the mandibular head [5]. However, some pathologies related to TMJ and surrounding areas may manifest as TMD in the earlier stages of RA independently of morphological changes and could be detected by conventional clinical examination. Therefore, it is important to identify TMJ disorders that are most frequent and specific to RA. It also seems important to determine if a more severe course of RA affects the condition of the TMJ and the occurrence of TMD. Many factors associated with RA may have a modifying effect on TMJ function and the development of TMD, forcing changes in the way we deal with them. It seems justified to indicate the most important features characterizing TMJ and muscle involvement in RA which formulates its clinical profile. For clinical purposes, more data are needed to better elucidate the relationship between the severity of RA, disease duration, and the severity of accompanying inflammation and TMJ condition to identify potential predictive factors for more severe TMD to occur. The aim of this study was to present a comprehensive assessment of TMJ involvement and TMD in RA patients and find possible correlations between selected factors associated with RA course and TMJ status.

## 2. Materials and Methods

The study comprised 40 patients diagnosed with RA (female/male ratio is 34:6, age range is 27–72 years) and 40 healthy subjects (HSs) who were included as healthy controls (female/male is ratio 21:19, age range is 34–58 years). All RA patients met RA diagnostic criteria according to the 2010 American–European Consensus Group (AECG) [7]. The healthy control group comprised healthy adult participants who signed patient’s consent. The healthy subjects (HSs) did not have any history of rheumatic disease. All RA patients were recruited consecutively from the Świętokrzyskie Rheumatology Centre in Saint Luke’s Specialist Hospital in Końskie, Poland and the Department of Rheumatology and Immunology Dietl Specialist Hospital in Cracow, Poland from January 2023 to June 2024. All HSs were recruited from the Dental Office Relax Dental Spa in Radom, Poland. The exclusion criteria included congenital craniofacial defects and temporomandibular joint defects, previous craniofacial injuries, previous orthognathic procedures, as well as patients treated in the past for temporomandibular joint disorders and undergoing intra-articular injections of TMJ. The patients who were wearing removable dentures were excluded from the study. Exclusion and inclusion criteria are presented in Table 1.

The patients’ comprehensive health assessment included a complete medical history and physical examination. Based on the medical history and clinical examination, we obtained demographic data, and data regarding duration of the disease, the type of treatment used, accompanying diseases, walking difficulties, circulatory and respiratory problems, and a number of swollen and involved joints. Joint pain was assessed using the Visual Analogue Scale (VAS) [8]. Disease activity was assessed using 28-Disease Activity Score (DAS28) [9]. DAS28 was divided into moderate and high level. Laboratory tests included rheumatoid factor (RF), erythrocyte sedimentation rate (ESR), white blood cell (WBC) count, platelet levels (PLTs), autoantibodies against citrullinated peptides (anti-CCP) detection, and C-reactive protein (CRP) value. The reference ranges for standard values at our laboratory were 4 × 10^3^–10 × 10^3^/mm^3^ for white blood count (WBC), less than 5 mg/L for C-reactive protein, and 1–10 mm/h for ESR.

The TMD were assessed following the standardized I and II Axis Research Diagnostic Criteria for Temporomandibular Disorders (RDC/TMDs) in the Polish version [10,11,12]. A single, skilled, nonblinded practitioner evaluated current symptoms and signs of both groups through an anamnestic questionnaire and clinical examination.

Sample size calculation was based on previous studies. Based on previous studies reporting 70% TMD prevalence in RA patients, we calculated that 38 participants per group would provide 80% of statistical power to detect a significant difference with α = 0.05, assuming 10% TMD prevalence in controls [13]. Additionally, for the results in which statistically significant differences were obtained, a power analysis of the applied tests was conducted. The power of test ranged from 0.70 to 1.

The protocol of this study was approved by the Institutional Review Board at Poznan University of Medical Sciences (number 672/22). Informed consent was obtained from all subjects before any study procedure was carried out. This study was performed in accordance with the ethical standards laid down in the appropriate version of the World Medical Association Declaration of Helsinki.

The calculations were carried out with Microsoft Excel 2016 and STATISTICA soft-ware (v.13 TIBCO, Palo Alto, CA, USA). Patients’ demographic data were analyzed using descriptive statistics. The Shapiro–Wilk test was used to assess the normal distribution of the variables. To examine the differences between the two groups, to verify if a normal distribution and equal variances were present, the *t*-test was used for unpaired samples; when non-equal variances were present, the Welch’s test was used. If normal distribution was not present, the Mann–Whitney U test was used. For qualitative variables, numbers (*n*) and proportions (%) were calculated and collected in cross tables. Categorical variables were presented in contingency tables and were analyzed using Chi^2^ Pearson’s test, Fisher–Freeman–Halton’ s test, and Fisher’s exact test. Spearman’s rank correlation analysis was used to find the correlations between the range of active and passive mouth opening, horizontal and vertical inter-incisal distance, number of TMD symptoms, and the selected disease activity markers such as CRP. For normally distributed data, results were presented as mean ± standard deviation (SD), and non-normally distributed data were expressed as median (interquartile range, IQR). Differences were considered statistically significant at *p* < 0.05.

## 3. Results

### 3.1. Demographic, Clinical, and Laboratory Characteristics of RA and HS Groups

The comprehensive demographic, clinical, and laboratory characteristics of the study participants are presented in Table 2.

### 3.2. Comparison of TMD in RA Patients and HS Group

There was a statistically significant difference in the prevalence of TMD between RA patients and healthy subjects. The predominant TMD symptom was orofacial pain. Orofacial pain was reported in 82.5% of RA patients and TMJ complains were reported in 75% of RA patients. Orofacial pain was more often reported bilaterally. Its intensity was statistically higher in RA patients in comparison to the HS group **(*p = 0.000*)**. Jaw movement induced TMJ pain or muscle pain. Maximum active and passive opening induced statistically significant orofacial pain in RA patients in comparison to the HS group **(*p = 0.000*)**. Active and passive mouth opening and protrusion induced mainly TMJ pain in RA patients. Muscle pain was less frequently induced during mouth opening and protrusion in RA patients. There were no differences in orofacial pain induced by laterotrusion between RA patients and healthy subjects. There was a difference in TMJ arthritis and TMJ arthralgia between RA patients and the HS group **(*p* *˂* *0.05*)**. The detailed data related to TMJ and muscle pain are presented in Table 3.

### 3.3. Assessment of Myofascial Pain and Muscles Involvement in RA Patients and HS Group

In 42.5% of RA patients, the pain on muscle palpation was reported. For all muscles examined through palpation independently on body sides, the reported pain was significantly higher in the RA group than in the HS group **(*p = 0.001*)**. There was a statistically significant difference in myofascial pain between RA and HS groups **(*˂0.05*)**. Moreover, there was no difference between *p value* and *p value* adjusted to age and gender. The most painful muscles in RA patients were the medial pterygoid muscles, the anterior belly of the digastric muscle, and the tendon of the temporal muscle. More muscles were painful in RA patients than in the HS group. Myofascial pain without limited mouth opening was found in 42.5% of RA patients. No participants from both the RA and HS groups had myofascial pain with limitation in mouth opening as presented in Table 4.

### 3.4. Self-Assessment of Orofacial Pain, Oral Health, and Degree of Somatization in RA and HS Groups

There was no difference in self-assessment of overall health between RA and HS groups (*p = 0.737*). Forty five percent of RA patients assessed their overall health as satisfactory. In turn, most of RA patients (60%) rated their oral health as good but statistically worse than healthy participants **(*p = 0.000*)**. There was a statistically significant difference in experiencing orofacial pain in the RA group compared to the HS group **(*p* *˂* *0.05*)**. In the RA group, 32 patients (80%) reported chronic orofacial pain. Most often, this pain was described as recurrent. Nineteen RA patients reported having orofacial pain for several years ranging from 1 to 8 years and fourteen RA patients reported having orofacial pain for several months ranging from 2 to 10 months. Fourteen RA patients (35%) sought help from various specialists to help with orofacial pain. Within the last six months, the most intensive orofacial pain was 7 on average and ranged from 3 to 8. In RA patients, orofacial pain was most often associated with a short-term but significant limitation in their functioning. Most often, orofacial pain in the RA group was classified as moderate. Moreover, RA patients more often complained of migraine headache and pain in the area adjacent to TMJ in comparison to healthy study participants.

Somatization degrees showed statistically significant differences in the presence of depression and non-specific physical symptoms, with both those related and unrelated to pain, between the control group and RA group. However, in-depth statistical analysis assessing which domain is dominant showed that in RA patients, non-specific symptoms including pain dominate over other domains (***p = 0.000***). In the control group, non-specific symptoms including pain dominate over those not including pain (***p = 0.023***). Among the disorders that contributed to depression in the RA group, the most severe were experiencing low energy or sluggishness, waking up early in the morning, restless sleep, the feeling of enormous exertion, and loss of interest in sexual activity. The detailed data related to orofacial pain and degree of somatization are presented in Table 5.

### 3.5. Assessment of TMD and Jaw Function Limitations in RA and HS Groups

Healthy participants reported no jaw function limitations contrary to RA patients (***p* *˂* *0.05*)**. In the RA group, difficulty chewing, yawning, eating hard food, and swallowing were reported most often in 75%, 70%, 65%, and 45%, respectively. The detailed distribution of jaw function limitations in the RA group is presented in Figure 1.

There was a statistically significant difference in TMD between RA and HS groups **(*p ˂* *0.05*)**. There were no TMDs in the HS group. The most frequent TMDs reported in the RA group were tinnitus and teeth grinding and clenching at night as well as during the day, followed by jaw clicks and crackles. A relatively large group of RA patients reported limited mouth opening and TMJ sounds such as friction and grinding. The detailed distribution of TMD reported in the RA group is presented in Figure 2.

### 3.6. Assessment of TMD and Orofacial Chronic Pain in the RA Group

There were no differences between TMD existence and the severity of orofacial chronic pain in RA patients. There were no correlations between the number of TMDs and jaw function limitations and the degree of orofacial pain. There was only correlation between the degree of orofacial chronic pain and the limitation in teeth brushing and face washing (***p* = 0.020**). The detailed data are presented in Table 6.

### 3.7. Assessment of TMJ Sounds in RA and HS Groups

Jaws movement induced TMJ sounds in RA patients. There was a statistically significant difference in TMJ sounds between RA patients and healthy subjects **(*p* *˂* *0.05*)**. In 52.5% of RA patients, clicking or crackles during jaw movement were found. The most frequent TMJ sounds in RA patients were crepitation. Most frequently, crepitations were described as slight. Most of the patients reported slight crepitation during jaw protrusion. The detailed distribution of TMJ sounds is presented in Table 7.

### 3.8. Assessment of Mandibular Kinematics and Movement Restriction in RA and HS Groups


**Mandibular kinematics and movement restriction in RA and HS groups are presents in Table 8.**


### 3.9. The Impact of RA Activity, Its Duration, and Severity of Inflammation on TMJ Function, TMJ Involvement, and Orofacial Pain

There was no statistically significant relationship between RA activity and chronic orofacial pain intensity and the degree of disability, (*p = 0.138*) and (*p = 0.075*), respectively. Although the number of TMD was higher in RA patients compared to the HS group, higher disease activity did not result in a statistically significant increase in the number of TMDs observed (*p = 0.407*). Moreover, there was no correlation between DAS and the incidence of any TMDs. DAS did not correlate with clicks or crackles (*p = 0.553*), friction or grinding sounds (*p = 0.130*), grinding or clenching at night (*p = 0.804*), and grinding and clenching during the day (*p = 0.884*). There was no correlation between DAS and morning jaw pain or stiffness (*p = 0.729*), and tinnitus (*p = 0.729*). Higher disease activity did not affect the limitation of jaw movements, including the range of maximum active and passive mouth opening, and vertical and horizontal inter-incisal distance, which were (*p = 0.440*), (*p = 0.530*), (*p = 0.795*), (*p = 0.709*), respectively. DAS did not produce TMJ arthralgia or arthritis, which were (*p* = 0.606) and (*p = 0.217*), respectively. There was no relationship between DAS and myofascial pain (*p* ˂ 1). Disease duration, and ESR and CRP levels did not correlate with the severity of chronic orofacial pain, which were (*p = 0.519*), (*p = 0.390*)*,* (*p = 0.717*)*,* respectively. Similarly, there was no correlation between the degree of disability and disease duration and the levels of ESR and CRP, which were (*p = 0.550*)*,* (*p = 0.153*) and (*p = 0.392*)*,* respectively. Disease duration and ESR level did not correlate with the number of the observed TMDs, which were (*p = 0.389*), (*p = 0.260*), respectively. In turn, CRP level negatively correlated with the number of TMD **(*r = −0.367, p = 0.019*)**. There were no correlations between disease duration, levels of ESR and CRP, and some of the TMDs such as grinding or clenching of teeth during the day and at night, TMJ noises, and tinnitus. There were no correlations between disease duration, ESR levels, and vertical ranges of motion in maximum active mouth opening, which were (*p = 0.767*) and (*p = 0.429*). In turn, CRP level negatively correlated with the range of maximum active mouth opening **(*r* *˂* *0, p = 0.042*)** (Figure 3).

There was no relationship between disease duration, ESR and CRP levels, and the range of maximum passive mouth opening, which were (*p = 0.524*), (*p = 0.825*), and (*p = 0.235*), respectively. There were no correlations between disease duration, ESR level, and horizontal inter-incisal distance, which were (*p = 0.282*) and (*p = 0.319*), respectively. In turn, there was a statistically significant positive correlation between horizontal inter-incisal distance and CRP level **(*r = 0.355, p = 0.024*)** (Figure 4).

There were no correlation between disease duration, levels of ESR and CRP, and the occurrence of arthritis or arthralgia of TMJ [(*p = 0.289*), (*p = 0.978*), and (*p = 0.635*), respectively], and between the disease duration, ESR and CRP levels, and occurrence of myofascial pain [(*p = 0.596*), (*p = 0.289*), and (*p = 0.484*), respectively].

There was no statistically significant difference in chronic pain intensity between RF positive (RF+) and RF negative (RF−) RA patients (*p* = 0.265). There was no correlation between RF seropositivity and degree of disability (*r = 0.179, p = 0.302*). There was no correlation between RF seropositivity and the number of TMDs (*r = 0.011, p = 0.942*). There were no correlations between RF seropositivity and the occurrence of clicks and crackles (*p = 0.684*), friction and grinding (*p = 0.079*), grinding and clenching at night (*p = 0.605*) and during the day (*p = 0.671*). There was a correlation between RF seropositivity and morning jaw pain or stiffness (***p = 0.030*)**. No correlation was found between RF and tinnitus (*p = 0.243*) and occlusal changes (*p = 0.281*). There were no correlations between RF seropositivity and ranges of maximal active and passive mouth opening [(*p = 0.907*) and (*p = 0.579*), respectively] and vertical and horizontal inter-incisional distances [(*p = 0.227*) and (*p = 0.912*), respectively]. No correlations were disclosed between RF and arthralgia (*p = 0.118*), and arthritis (*p = 0.176*), and myofascial pain (*p = 0.520*).

There were no correlations between anti-CCP seropositivity and orofacial pain intensity, degree of disability, number of TMDs, TMJ sounds, and symptoms of bruxism. There were no differences in the ranges of maximal active and passive mouth opening between anti-CCP positive (anti-CCP+) and anti-CCP negative (anti-CCP−) RA patients. There was a statistically significant difference in horizontal inter-incisional distance between anti-CCP+ and anti-CCP- patients (***p = 0.018***). There were no correlations between anti-CCP seropositivity and the occurrence of arthralgia and arthritis and myofascial pain. The number of swollen joints and of painful joints among RA patients did not affect orofacial pain intensity, TMD occurrence, arthritis, and arthralgia. There were no statistically significant correlations between the number of painful and swollen joints and any TMJ involvement and limitations in jaws functioning.

## 4. Discussion

The presented study found a very high prevalence of TMD in patients with RA, indicating that RA could be a predisposing factor for them to develop. These findings are similar to those obtained by Mustafa et al. who revealed that 70% of RA patients suffered from some degree of TMD, and Cordeiro et al. who revealed that 75% of RA patients complained of orofacial pain as a main TMD [13,14]. In our study, TMDs were poorly RA-specific, quite subjective, and some of them remained clinically silent. Some were revealed on clinical examination, palpation, or during jaw movement. Furthermore, most of the observed symptoms were widespread, involving many muscular and fascial points adjacent to TMJ. These observations are consistent with the findings obtained by Sadura–Sieklucka et al. who indicated that most of RA-related TMD are subjective sensations and could manifest during TMJ palpation [5]. Although they were quite common in RA patients, their intensity was relatively mild. However, there are no consistent findings related to the severity of TMD in RA patients [2,13,15]. Some results indicated that more than half of the patients had severe TMD, presenting with debilitating symptoms or with a significant degree of bony destruction [2]. However, according to some previous studies, the majority of RA patients suffered from mild TMD in approximately 39.5%, followed by moderate TMD (24.7%). Only 6% of RA patients demonstrated severe TMD [14]. In the study conducted by Lin et al., 91% of RA-examined patients had a subjective TMD score ˂ 6 [2]. Therefore, most RA patients do not see any direct connection of their TMD with RA course and seek help in resolving TMJ problems. One of the possible explanations for this discrepancy is that TMJ discomfort or complaints are likely to pose a lesser problem for a patient with their joint pains in other parts of the body. Moreover, most TMJ complaints observed in RA patients are not associated with severe limitations in jaw function. These functional limitations might be easily reduced by avoiding routine daily jaw activities such as eating hard foods. In our study, there was no one specific TMD associated with RA. It is hard to indicate one obvious RA-specific TMD. The range of TMD detected in the RA group is similar to the TMJ disorders observed in the general population. In the overall population, quite a big number of abnormalities are found during TMJ clinical examinations. Therefore, for some patients in our study, TMD might not only result from RA. Moreover, previous studies have indicated some potential risk factors for developing TMD, including being of the female gender, increased body weight, younger age, insomnia, stroke, and mental disorders [13,16,17]. However, high prevalence of TMD in RA patients allows us to treat RA as an independent risk factor for the development of TMD. It is consistent with other studies which report that the prevalence of TMD in the RA population is 2.5 higher than in a healthy group [17]. Furthermore, the same authors paid attention to the impact of hypertension, stroke, and the effectiveness of RA therapy on the development of TMDs in RA patients [17]. Some drugs, including disease-modifying antirheumatics, may inhibit the occurrence of TMD in RA [13,16,17].

Our findings revealed that chronic orofacial pain is a predominant symptom in RA patients in comparison to the healthy group. This pain may occur independently of the disease stage, disease activity, and severity of the accompanying inflammation. Its intensity did not correlate with inflammatory parameters. Moreover, it was recurrent in nature and occurred both at rest and during jaw movements. It is not only defined as arthralgia, but could be classified as pain sensations involving TMJ structures and also adjacent muscular and fascial areas. This orofacial pain could be classified as moderate. Fortunately, it was not associated with relatively long-standing disability and handicap in routine daily activities. Our results are consistent with other studies that indicated orofacial pain as the most common TMD in RA patients [18,19,20]. Patients with RA and other autoimmune rheumatic diseases suffer from muscle and TMJ pain during palpation more often in comparison to healthy subjects [18]. Similarly to the results obtained in our study, other researchers found that RA patients quite often report pain in the masticatory muscles, headache, and restricted mouth opening [19]. Headache was observed in 58% of RA patients and masticatory muscles pain was observed in 57% of RA patients [19]. Although, in our study, there was no statistically significant difference in the headache occurrence between RA and HS groups, it seems that RA patients are prone to headache. Moreover, pain sensations in RA patients could arise from other adjacent areas, including the TMJ, masticatory muscles, and shoulders. These pain experiences might overlap and together give a sensation described as orofacial pain. Higher prevalence of migraine headache in the RA group could be associated with cervical spine and shoulder involvement often found in RA patients. One of the frequent pain sensations observed in RA patients is arthralgia. In our study, 75% of RA patients reported some subjective TMJ complaints. This pain was not associated with severe limitations in jaw movement and their reduced function. Pain intensity ranged from mild to moderate. One of the possible explanations for this discrepancy is the unique structure of TMJ in comparison to other involved joints. Retrodiscal tissue is rich in blood supply and may act as an effective drainage system for joint exudates. This mechanism may reduce TMJ pain and prevent severe inflammation. The same mechanism may explain a relatively small number of RA patients who demonstrated symptoms of TMJ arthritis. Another type of pain sensation found in the RA group was muscle tenderness. All extraorally and intraorally examined muscles were painful. These results are consistent with other studies where muscle pain was reported in RA patients, with 53.29% of patients reporting pain in the masseter muscle, 29.6% of patients reporting pain in the temporal muscle, and 40% of patients reporting pain in the pterygoid muscles [20]. Similar results were obtained by Ardic et al. who revealed that 55% of RA patients demonstrated myofascial pain [21]. According to the same authors, pain intensity ranged from gentle tenderness to muscle spasm [20]. In the study of Witulski et al., palpation tenderness of the muscles was detected in as many as 93% of the patients with RA [22]. The high prevalence of myofascial pain suggests mechanisms beyond synovitis, including direct synovial inflammation, central sensitization due to chronic pain, altered biomechanics of jaw movement, and systemic inflammatory mediators affecting the muscle tissue [14]. The most painful lateral pterygoid muscles in RA included all parts of the temporal muscle and its tendon, suboccipital muscles, lateral pterygoid muscle, and superficial and deep parts of the masseter muscles [22]. One of the possible explanations of so high prevalence of muscle pain could be predisposition to myalgias, fibrositis, and ligament involvement in RA. Furthermore, some muscle pain sensations were induced by jaw movement. In some cases, jaw movement could trigger joint pain and in other cases, the same movement could cause either muscle pain or combined joint and muscle pain. These schemes could be associated with the radiation of pain and RA progression toward joint and muscle involvement. It is estimated that TMDs are related to 7.5 times higher odds for the presence of orofacial pain [23]. Muscle pain could be an indicator of the extent of dysfunction and the pathology of the masticatory system. The observation of medial pterygoid muscles and temporalis tendons showing the highest pain scores is consistent with their stabilizing role in mastication, potentially explaining their vulnerability [23]. Another explanation for the high prevalence of myofascial pain and muscle soreness in RA patients compared to the control group is a possible correlation between myofascial and orofacial pain and bruxism symptoms, including teeth clenching and grinding observed in RA patients. These symptoms may induce myofascial pain. It is hard to definitively understand and decide if myofascial pain is a primary symptom and more RA-specific than teeth grinding or clenching. Moreover, it seems that myofascial and orofacial pain can result from depression, which is quite frequent among RA patients. High degree of disability accompanying RA may induce persistent stress, leading to increased tension of masticatory muscles and the development of myofascial pain. In turn, except TMD, RA itself is responsible for muscle pain. Therefore, we can treat RA as an important factor contributing to muscle and orofacial pain. In the study conducted by Mesic et al., RA was related to 3.4 times higher odds for orofacial pain compared to healthy subjects [23]. Combination of arthralgia and an age ≥ 55 years were main predictors of RA. Moreover, RA patients were characterized by higher chronic pain grade and pain intensity in comparison to the control group. Chronic pain observed in the RA group was associated with increased degree of somatization and depression [23,24]. It was estimated that half of RA patients who had clinical TMD developed both psychological depression and nonspecific physical symptoms. Furthermore, these physical symptoms had a tendency to accompany the disease with no obvious physical causes [24].

Apart from various forms of pain sensations affecting the head and neck area, it seems that RA is a predisposing factor for the development of other TMDs. One of the most often reported jaw function limitations in the RA group is chewing difficulties, especially of hard food. Some of the reduced functions accompanying RA might directly result in the occurrence of these TMDs. On the other hand, RA patients are prone to severe periodontal diseases and teeth mobility. Radwan–Oczko et al. indicated a strong relationship between RA and oral health status parameters, including poor gingival state, gingival bleeding, and difficulties in biting or chewing. Furthermore, higher DAS correlated with more severe difficulties in biting or chewing, discomfort or pain in the oral cavity, a sensation of movable teeth, and gingival bleeding, which are important predictors of periodontal disease [25,26]. Patients with RA demonstrated oral hypofunction in comparison to the general older population. This dysfunction manifested as the higher prevalence of chewing difficulties, difficulties in eating tough foods, swallowing difficulties, and dry mouth among older RA patients [27]. Similar distributions of the jaw function limitations were found in the study conducted by Bessa-Nogueira et al., who indicated that the most frequently seen problems for RA patients were eating hard food (50.8%), chewing (39.3%), yawning (34.4%), and smiling or laughing (24.6%) [28]. Moreover, the prevalence of these symptoms corelated with some clinical and laboratory parameters used to evaluate RA severity, such as the number of painful joints, the Health Assessment Questionnaire (HAQ) score, and RF positivity [28]. In our study, a relatively large number of RA patients demonstrated tinnitus in comparison to healthy controls. Previous studies indicated tinnitus as a frequent TMD in RA patients [5,29]. Rheumatoid arthritis as an autoimmune disease is responsible for the deposition of immune complex in cochlea hair and subsequent sensorineural hearing loss or neuritis. A possible source of tinnitus in RA is the ototoxic action of some drugs used in RA treatment, such as steroids, nonsteroidal anti-inflammatory drugs, and disease-modifying antirheumatic drugs [29].

Another TMD often found in RA patients compared to healthy controls is TMJ sounds. The predominant TMJ sounds are slight crepitations. Similar results were obtained by Bessa-Nogueira et al. who found sounds on TMJ movement in 49% of RA patients with a click in 19.7% and crepitation in 29.5% [28]. However, a lack of radiological assessment of TMJ in our study makes it more difficult to decide on TMJ sounds as an RA-specific TMD. Crepitation could be treated as a reliable predictor of erosion [19,28]. Moreover, some studies pointed out that TMJ sounds might correlate with RA duration and the severity of inflammation. TMJ crepitus is associated with an early stage of RA when TMJ cartilage and bone predominate and are correlated with TMJ pain intensity [30,31]. Therefore, early and effective therapy against RA and reduction in the accompanying inflammation play a crucial role in preventing future TMJ degeneration. Some researchers revealed that TMJ clicking could be observed in RA as well as healthy controls. On the other hand, TMJ crepitus is more specific to RA because it is found preponderantly in RA patients as opposed to healthy subjects [22]. Rheumatoid arthritis is a chronic inflammatory disease that could be a predisposing factor for degenerative diseases, such as osteoarthritis. With a long disease duration, these two processes can overlap and induce TMJ sounds.

In our study, there was a significant difference in the maximal range of active opening between RA patients and HS. This finding was also present in other previous studies [19,20,22,28]. Impaired mouth opening was found in 42% of RA patients and reduced mouth opening of less than 40 mm was detected in 71% of RA patients. Reduced mouth opening was significantly more frequent in RA patients than in the control group [19]. Contrary results were obtained by Jalal et al., who did not show any differences in the mean unassisted mouth opening between RA patients and the controls [32]. However, it is quite hard to indicate real causes of these limitations. These reduced ranges of mandible movement may result from intra-articular and muscular factors, including reduced capacity for extension of the capsular ligament, difficulties in translation of the head of the mandible by anterior disc displacement, formation of intra-articular fibrous adhesions, or elevator muscles shortening by muscle contracture or inflammation [28]. Furthermore, some researchers indicated that lower maximum mouth opening capacity is associated with the established stage of RA and is more severe with longer disease duration [30]. Other possible causes of decreased range of motion in RA patients are reduced joint space, sclerosis, or changes in mandible head as an adaptive mechanism.

Generally, patients with longer disease duration have more clinical symptoms of TMJ [32]. However, there is no consensus as to the relationship between the occurrence of TMD and disease duration [2,13,32]. Unexpectedly, our results found no correlation between TMD occurrence and the severity and duration of RA. This paradox may be explained by early inflammatory damage, modern DMARD/biologic therapies preventing progression, individual susceptibility factors such as genetic background or craniofacial anatomy, and adaptive mechanisms maintaining function despite structural changes [18,28]. Previous studies reported that some TMDs might develop in the early stage of RA even within 1 year after the onset of the generalized disease. These symptoms rarely appear before the generalized disease. However, the largest proportion of RA patients develop TMJ symptoms later, even more than 5 years after the onset of the disease [2,21,30,32]. Thus, continuous monitoring of TMD during the course of RA should be mandatory. There is no consensus about the relationship between the frequency of some TMDs in RA and the disease duration as some TMDs may depend on disease duration, and some others may not. According to Jalal et al., there was no significant difference in jaw functions, especially in the range of active and passive mouth opening, lateral excursion, and mandible protrusion between RA patients with 1–5 years of disease duration and 6–10 years of disease duration [32]. On the other hand, orofacial pain intensity and disease duration were correlated. Patients with longer disease duration have more clinical symptoms of TMD, such as jaw lock, facial pain, and jaws clicking during mouth opening [32]. TMJ involvement was more frequent in RA patients with longer disease duration [32]. In turn, Kroese et al. observed the evolution of TMJ complaints in RA with disease progression. Early RA, up to 2 years, is associated with higher general pain and TMJ pain intensity at rest and during maximum mouth opening, early TMJ cartilage degradation, and TMJ crepitus. On the other hand, established RA may induce reduction in TMJ function. Early RA results in TMJ pain without severe limitation in jaw function and a relatively small number of painful muscles. With the disease duration established, RA triggers a more severe reduction in jaw function and a larger number of painful joints. This is likely because the number of involved joints increases with disease duration and is preceded by increased systemic inflammatory activity [30]. Both general and TMJ pain intensity seem to be higher in early RA when systemic inflammatory activity is usually higher. It seems that severity of TMD in RA patients may not be directly related to disease duration, but rather to the rate of disease progression and the duration of active inflammation. Long duration of active inflammation may result in the articular damage to TMJ, rather than long disease duration. Furthermore, latent TMD may be predisposed to the earlier development of TMJ dysfunction during the course of RA. Early, unidentified TMD before the onset of RA may make them more severe in RA patients. In addition, the effectiveness of RA treatment may modify the duration and severity of accompanying TMD.

Another finding obtained in the study is that there is no direct correlation between RA severity and the occurrence of TMD. Previous studies are inconsistent regarding the relationship between RA activity and TMJ involvement [2,13,19,33,34,35]. According to Ahmed et al., higher DAS correlated with TMJ pain during chewing and maximum mouth opening, as well as the number of painful regions [33]. More objective findings were obtained by Bono et al. who did not note any correlations between DAS and radiological changes in TMJ in RA patients [19]. This inconsistence may arise from the different parameters used to assess RA severity in various disease stages. Most commonly used parameters included the number of swollen joints and tender joints, the peak of ESR, RF and CRP, the Physician’s Global Assessment score, score of hand-bone erosion, score of hand-joint space narrowing, and total hand-joint destruction score [2]. According to Lin et al., the number of swollen joints and painful joints made fewer sensitive parameters that did not correlate with TMD severity in RA patients [2]. In turn, the number of swollen and tender joints could be associated with more severe disability and impaired life quality [20]. In our opinion, difficulties in identifying one single parameter correlating with the advancement of changes in TMJ are the result of their different diagnostic values and universality for different stages of RA. The number of tender or swollen joints is more useful in monitoring the disease progression in initial RA stages. On the other hand, joint space narrowing usually appears later in a more advanced RA stage. Furthermore, DAS 28 does not consider functions of the stomatognathic system, such as chewing, yawning, and speaking. Therefore, it does not seem to be an ideal parameter to evaluate TMD and TMJ involvement in RA patients. This feature could explain no correlations found in our study between TMD occurrence and RA severity.

Laboratory parameters such as CRP and ESR that reflect inflammation may be important factors to assess TMJ involvement in RA patients. Their increased levels may predict clinical symptoms [32]. In our study, a significant correlation has been found between the range of active mouth opening and CRP value in RA patients. Increased CRP level correlated with reduced active mouth opening. This relationship highlights the importance of an active inflammatory process as a potential factor to determine limited TMJ function. ESR tends to be an indicator of chronic inflammation while CRP is more useful to assess acute inflammation; it declines rapidly after the inflammation is controlled. It seems that chronic inflammation reflected by increased ESR is associated with overall grade of progression of radiological changes in TMJ. In turn, raised levels of CRP are associated with TMJ bone loss and significant functional limitation [36]. Therefore, reducing acute inflammation plays a crucial role in maintaining optimal TMJ function. Moreover, CRP values in previous studies correlated with radiological symptoms, showing progression of TMJ involvement in RA patients [21,36]. Contrary to these findings, Lin et al. found that ESR value correlated with TMD severity in RA patients and, in turn, CRP did not correlate with TMD severity [2]. Jalal et al. indicated that ESR is a more universal parameter than CRP because increases in the level of ESR correlated with all clinical symptoms of TMJ while CRP level showed positive correlation with all symptoms except joint clicking [32]. Both ESR and CRP are non-specific for RA and are not ideal markers for the assessment of TMJ involvement. Some previous studies showed considerable individual variations between the levels of CRP and ESR and the progression of radiological changes in TMJ. However, CRP is more a valuable marker for the monitoring of radiological TMJ changes than ESR in patients with well-controlled RA because the progression of radiographic changes that occurs in the TMJ of patients with well-controlled RA during a period of 25–46 months could be related to increased CRP levels [37]. In our study, the increased value of CRP negatively correlated with the number of reported TMDs. A possible explanation of this finding is that subjective TMD might be masked by objective limitations in TMJ function, especially the range of active mouth opening. Limitation of objective mouth opening may result in the avoiding some routine jaw functions. Previous studies did not show any correlations between CRP value and TMD occurrence [13]. Increased CRP levels were not correlated with pain and painful forms of TMD [38]. From a clinical perspective, our findings emphasize the importance of TMD screening in all RA patients, regardless of disease duration [2,5]. Monitoring of CRP levels may provide prognostic value, as elevated CRP is associated with greater functional limitations [32]. Early and targeted control of systemic inflammation could help to preserve TMJ function and effective management requires multidisciplinary collaboration between rheumatologists, dentists, oral medicine specialists, and physiotherapists [6].

The strengths of this study include the standardized use of RDC/TMD diagnostic criteria [10,11], comprehensive assessment of clinical features, and correlation with systemic inflammatory markers. However, limitations must be acknowledged. This study is mainly limited by the small sample size. While our sample size was sufficient for the primary analyses, larger cohorts are required for subgroup analyses. The obtained findings should be verified on a larger and more homogenous RA population. Moreover, the cross-sectional design precludes causal inference. The absence of examiner blinding may have introduced assessment bias, although standardized diagnostic protocols mitigate this risk. Demographic differences, despite statistical adjustment, may have influenced outcomes. Furthermore, medication effects were not fully accounted for. The RA patients enrolled in the study group were treated with different medicines including anti-inflammatory drugs. These drugs may change levels of ESR and CRP. Except anti-inflammatory therapies, some patients were administered biological treatment. The effectiveness of RA therapy may have influenced TMJ involvement. Therefore, current and previous RA therapies, as well as co-existing diseases and therapies, may have interfered with the obtained results. A significant discrepancy in disease duration can have a significant impact on TMJ involvement. Due to a significant delay in implementing RA therapy after the first disease symptoms have occurred, we included in the study not only the patients in the initial stage of the disease. Another potential limitation of this study is that there was no assessment of the patients’ oral health. There is a direct relationship between TMJ function and general oral health. Missing teeth may increase the risk of TMD and induce progressive changes in TMJ. Some obtained findings are based on the questionnaire. For a comprehensive assessment of TMJ involvement in RA, these results should be correlated with radiological examination. A lack of any radiological examination, including MRI, makes it impossible to find correlations between the TMD findings and real TMJ arthritis. This limitation of the study does not allow us to determine the real cause and nature of the pain sensation and its connection with the active TMJ inflammation. Another limitation of the study is the age and gender discrepancy between HS and RA patients. Previous studies revealed significant differences in prevalences of arthralgia, myalgia, headache attributed to TMDs and disc displacements between young adults and older people as well as between males and females. Younger patients presented more intra-articular conditions and fewer pain-related conditions than the older generations. Furthermore, the female gender is predisposed to degenerative joint diseases. Being a female of advanced age is associated with the higher risk of painful TMD [39,40]. On the other hand, some of these findings may be attributed to differences in social stress, emotional responses, general and mental health pain attitudes, and help-seeking behaviours [40].

Future research should prioritize longitudinal designs with imaging correlation to capture the natural history of TMJ involvement [22,28]. Studies investigating inflammatory mediators and biomarkers may help identify specific pathways of TMJ involvement [41]. Clinical trials testing targeted interventions, genetic studies exploring susceptibility factors, and quality-of-life analyses would further clarify disease burden and management strategies [23,25].

## 5. Conclusions

RA patients are at risk of developing various TMDs, regardless of disease duration, age, and gender. They have a diverse clinical presentation, involving structures adjacent to TMJ. Predominant TMDs are chronic orofacial pain and limitations in jaw mobility, mainly in the range of maximum mouth opening, which significantly impact daily activities such as chewing, yawning, swallowing, and eating hard foods. RA should be regarded as an important factor to produce future degenerative changes in TMJ, including TMJ osteoarthritis. As there is no correlation between disease duration and TMD occurrence, constant monitoring of RA patients and implementing early treatment of any TMDs are needed. A crucial role in the treatment of TMD in RA patients is fighting an active inflammation accompanying RA. The observed link between active inflammation, reflected by elevated CRP and reduced jaw function suggests that optimal control of systemic inflammation may preserve TMJ function. Both articular and myofascial mechanisms contribute to orofacial pain, necessitating comprehensive assessment and multimodal management. Standard disease activity measures such as DAS28 and disease duration are poor predictors of TMD, highlighting the need for TMJ-specific assessment tools. These findings support integration of TMD screening into RA care protocols. Early recognition and multidisciplinary management may improve quality of life and prevent progressive TMJ damage. Future longitudinal studies with imaging correlation are warranted to clarify the natural course of TMJ involvement in RA and to develop targeted therapeutic strategies.

## Figures and Tables

**Figure 1 jcm-14-07381-f001:**
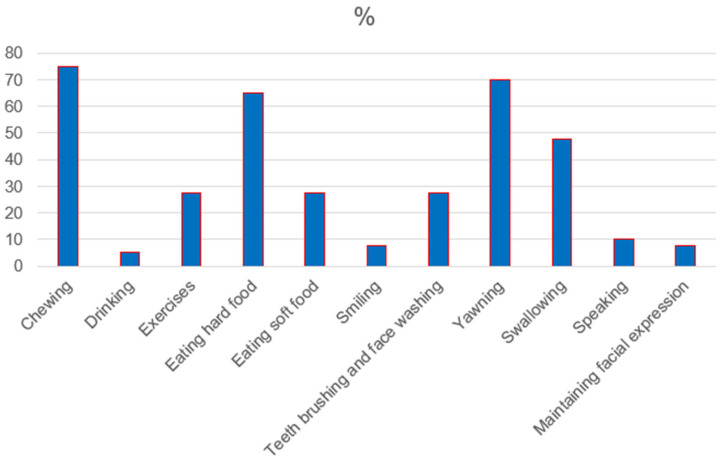
Distribution of jaw function limitations in RA group.

**Figure 2 jcm-14-07381-f002:**
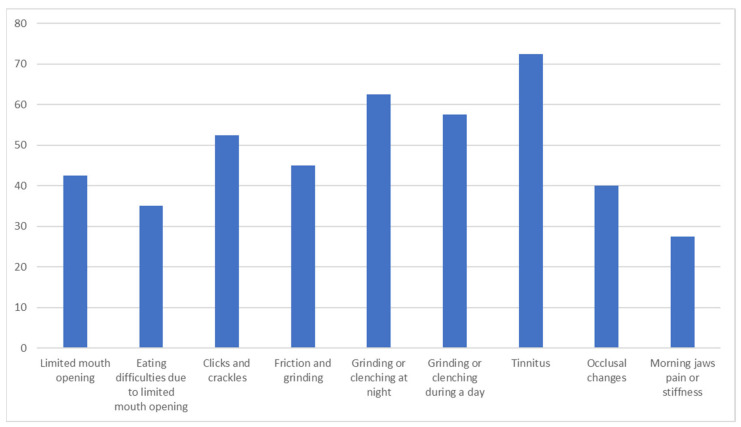
Distribution of TMD in RA patients.

**Figure 3 jcm-14-07381-f003:**
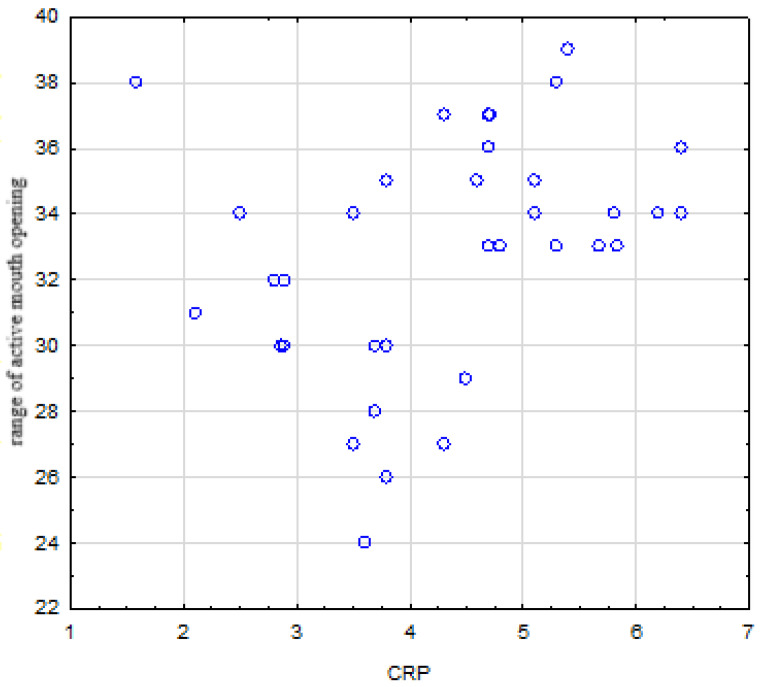
Negative correlation between CRP level and range of active mouth opening.

**Figure 4 jcm-14-07381-f004:**
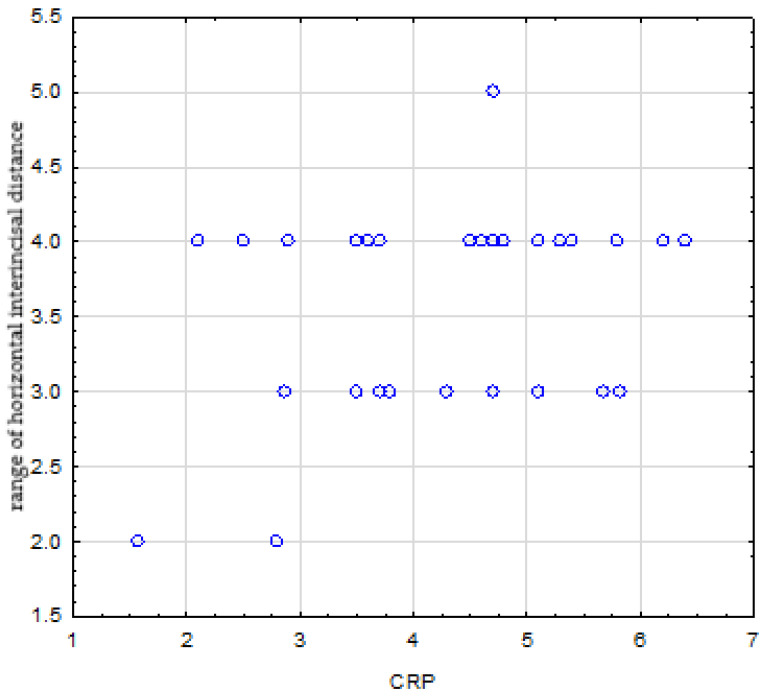
Positive correlation between CRP level and range of horizontal inter-incisal distance.

**Table 1 jcm-14-07381-t001:** Inclusion and exclusion criteria.

Inclusion Criteria RA Group	Inclusion Criteria HS Group	Exclusion Criteria Both Groups
Adults aged ≥ 18 years	Adults aged ≥ 18 years	Congenital craniofacial defects
Written informed consent	Written informed consent	Congenital temporomandibular joint defects
Confirmed RA diagnosis according to 2010 ACR/EULAR classification criteria [7]		Previous craniofacial injuries
Ability to understand and complete questionnaires	Ability to understand and complete questionnaires	Previous therapy of temporomandibular joint disorders
	No history of rheumatic disease	History of orthognathic procedures
	No current TMJ complaints	Current or previous TMD treatment including intra-articular injections Wearing removable dentures
		Pregnancy or lactation

**Table 2 jcm-14-07381-t002:** Demographic, clinical, and laboratory characteristics of RA and healthy subject (HS) groups.

Parameters	Healthy Subjects (HSs) *n* = 40	RA Patients *n* = 40	*p Value*	*p Value* *Adjusted to Age and Gender*
Age			***0.002* ^1^**	
Mean ± SD	46 ± 5.75	52.5 ± 11.5		
Median (ranges), years	46 (34–58)	53 (27–72)		
Gender female/male, *n*	*21/19*	*34/6*	***0.001* ^2^**	
Disease duration years		6 (12.5)		
Ranges		0.5–40		
Disease diagnosis ≤1 years, *n* (%)		*8* (20)		
Treatment duration years, median (IQR)		5 (9)		
Ranges		0.5–38		
Treatment				
Nonsteroidal anti-inflammatory drugs, *n* (%)		*32* (80)		
Corticosteroids, *n* (%)		*16* (40)		
Disease-modifying antirheumatic drugs (DMARDs), *n* (%)		*24* (60)		
Biological treatment, *n* (%)		*5* (12.5)		
Number of biological treatment cycles				
1 cycle, *n*		*2*		
2 cycles, *n*		*4*		
Laboratory tests				
ESR	11.4 (1.2)	14.5 (6.5)	***0.000* ^3^**	***0.001* ^3^**
CRP	1.5 (1.7)	4.5 (1.8)	***0.000* ^3^**	***0.000* ^3^**
WBC	7.3 (3.2)	11.3 (1.8)	***0.000* ^3^**	***0.000* ^3^**
PLT	152 (15.5)	208 (73)	***0.000* ^3^**	***0.000* ^3^**
RF	1.25 (1.25)	226.5 (63.6)	***0.000* ^3^**	
Anti- CCP				
+ positive, *n*		*30*		
− negative, *n*		*10*		
Accompanying diseases, *n*:	*0*	*25*		
Osteoporosis		*9*		
Inflammatory bowel disease		*7*		
Dry eye syndrome		*2*		
Walking difficulties, *n*	*0*	*23*		
Circulatory and respiratory problems, *n:*	*0*	*13*		
Hypertension		*8*		
Heart rhythm disturbances		*2*		
Interstitial lung disease		*3*		
Chronic obstructive pulmonary disease		*1*		
Family history of rheumatic diseases, *n*	*11*	*11*	*1.000* ^2^	
DAS 28		5 (0.6)		
Disease activity, *n*:				
Moderate (3.2–5.1)		*23*		
High (>5.1)		*17*		
VAS (0–10)		7 (2)		
Number of tender joints		8 (4)		
Number of swollen joints		9 (4.5)		

Unless otherwise stated, the data are expressed as medians (IQR, interquartile range); *n*, number; RA, rheumatoid arthritis; SD, standard deviation; ESR, erythrocyte sedimentation rate; CRP, C-reactive protein; WBC, white blood cells; PLT, platelets; RF, rheumatoid factor; anti-CCP, autoantibodies against citrullinated peptides; DAS, Disease Activity Score; VAS, Visual Analogue Scale; ^1^, Welch’s test; ^2^, Chi^2^ Pearson’s test; ^3^, Mann–Whitney U test.

**Table 3 jcm-14-07381-t003:** TMD symptoms in RA patients and healthy controls.

	Healthy Subjects *n* = 40	RA Patients *n* = 40	*p* Value	*p Value Adjusted to Age and Gender*
**TMJ complaints, *n* (%)**	*0* (0)	*30* (75)		
**Total orofacial pain prevalence, *n* (%)**	*0* (0)	*33* (82.5)		
**Orofacial pain on palpation, *n*:**				
**Bilateral**	*0*	*18*		
**Unilateral**	*0*	*15*		
Right side (RS)	*0*	*10*		
Left side (L)	*0*	*5*		
**Orofacial pain on palpation RS/LS, *n*:**				
TMJ pain	*0*	*14/11*		
Muscle pain	*0*	*6/4*		
Both TMJ and muscle pain	*0*	*11/9*		
**TMJ pain intensity on palpation:**				
**Right TMJ:**				
Lateral part	1 (1)	3 (1)	***0.000* ^1^**	***0.000* ^1^**
Posterior part	1 (1)	3 (1)	***0.000* ^1^**	***0.000* ^1^**
**Left TMJ:**				
Lateral part	1 (1)	2 (1)	***0.000* ^1^**	***0.002* ^1^**
Posterior part	1 (1)	2 (1)	***0.000* ^1^**	***0.000* ^1^**
**Orofacial pain during jaw movement on Right/Left Side RS/LS, *n*:**				
during maximum active opening				
TMJ pain:	*0/0*	*19/17*		
Muscle pain	*0/0*	*9/12*		
Both TMJ and muscle pain	*0/0*	*6/5*		
during maximum passive opening:				
TMJ pain	*0/0*	*19/15*		
Muscle pain	*0/0*	*11/12*		
Both TMJ and muscle pain	*0/0*	*3/0*	*0.190* ^2^/*0.095* ^2^	*0.255* ^2^/*0.477* ^2^
Right laterotrusion, *n:*				
TMJ pain	*3/2*	*9/6*		
Muscle pain	*20/20*	*21/24*	*0.198* ^3^/***0.003* ^2^**	*0.108* ^3^/*0.311* ^2^
Both TMJ and muscle pain	*5/6*	*3/1*		
Left laterotrusion, *n*:				
TMJ pain	*6/6*	*8/6*	***0.000* ^3^**/*0.068* ^3^	***0.006* ^3^**/*0.527* ^3^
Muscle pain	*18/14*	*23/28*		
Both TMJ and muscle pain	*6/5*	*6/3*		
Protrusion, *n*:				
TMJ pain	*4/7*	*20/17*		
Muscle pain	*17/15*	*8/11*		
Both TMJ and muscle pain	*7/8*	*9/3*		
Arthritis of TMJ, *n*				
Right TMJ	*0*	*1*		
Left TMJ	*0*	*19*		
Arthralgia of TMJ, *n*				
Right TMJ	*0*	*24*		
Left TMJ	*0*	*0*		

Unless otherwise stated, the data are expressed as medians (IQR interquartile range); ^1^, Mann–Whitney’s U test; ^2^, Fisher–Freeman–Halton’s test; ^3^, Chi^2^ Pearson’s test.

**Table 4 jcm-14-07381-t004:** Myofascial pain and muscle involvement in RA patients and HS group.

	Healthy Subjects *n* = 40		RA Patients *n* = 40		*p* Value		*p Value Adjusted to Age and Gender*
	Right Side RS	Left Side LS	Right Side RS	Left Side LS	Right Side RS	Left Side LS	
**Pain on muscles palpation, *n* (%)**	*0* (0)		*17* (42.5)				
**Extraoral muscles and neck muscles:**							
Temporal muscle							
Anterior	2 (2)	1 (1.5)	3 (1)	2 (1)	***0.000* ^1^**	***0.000* ^1^**	** *˂0.05* **
Central	1 (1)	1 (1)	2 (0)	2 (0)	***0.000* ^1^**	***0.000* ^1^**	** *˂0.05* **
Posterior	1 (1)	1 (0.5)	2 (0)	2 (0)	***0.000* ^1^**	***0.000* ^1^**	** *˂0.05* **
Masseter muscle							
Superior	1.5(1)	1 (1)	2.5(1)	2(0.5)	***0.000* ^1^**	***0.000* ^1^**	** *˂0.05* **
Central	1 (0)	1 (0)	3 (1)	2 (1)	***0.000* ^1^**	***0.000* ^1^**	** *˂0.05* **
Inferior	1(0.5)	1 (1)	2(0.5)	2 (0)	***0.000* ^1^**	***0.000* ^1^**	** *˂0.05* **
Stylomandibular muscle and posterior belly of digastric muscle	1 (1)	1 (1)	3 (1)	2 (1)	***0.000* ^1^**	***0.000* ^1^**	** *˂0.05* **
Med pterygoid muscles and anterior belly of digastric muscle	1 (1)	1 (1)	3 (1)	3 (1)	***0.000* ^1^**	***0.000* ^1^**	** *˂0.05* **
**Intraoral muscles:**							
Lat pterygoid muscles	1(0.5)	1 (0)	2 (1)	2 (1)	***0.000* ^1^**	***0.000* ^1^**	** *˂0.05* **
Tendon of temporal muscle	2 (2)	1 (2)	3 (0)	3 (1)	***0.000* ^1^**	***0.002* ^1^**	** *˂0.05* **
**Total number of painful muscles, *n* (%)*:***							
Ranges	*9–20* (45–100)		*14–20* (70–100)				
**Myofascial pain, *n* (%):**	*0*		*17* (42.5)				
**Myofascial pain with limited mouth opening, *n* (%)*:***	*0* (0)		*0* (0)				

Unless otherwise stated, the data are expressed as medians (IQR interquartile range); RA, rheumatoid arthritis; *n*, number; %, percentage; ^1^, Mann–Whitney’s U test; Lat, lateral; Med, medial.

**Table 5 jcm-14-07381-t005:** Comparison of orofacial pain and degree of somatization between RA and HS group.

	HS *n* = 40	RA Patients *n* = 40	*p Value*	*p Value Adjusted to Age and Gender*
Overall health	4 (0)	4 (2)	*0.737* ^1^	*0.996* ^1^
Oral health	4 (1)	3 (1)	***0.000* ^1^**	***0.006* ^1^**
Pain or swelling of joints excluding TMJ, *n*	*0*	*11*		
**Orofacial pain**				
Chronic orofacial pain, *n* (%)	*0* (0)	*32* (80)		
Orofacial pain characteristics, *n*:				
Single incident	*0*	*6*		
Recurrent	*0*	*23*		
Constant	*0*	*3*		
Chronic orofacial pain intensity		7 (1)		
Chronic orofacial pain duration, years		2 (2.3)		
Number of days with impaired mobility				
Ranges:		3–10 days		
0–6 days, *n*		*23*		
7–14 days, *n*		*17*		
15–30 days, *n*		*0*		
≥31 days, *n*		*0*		
Pain impact on everyday life, *n:*				
0		*2*		
1		*1*		
2		*20*		
3		*12*		
Handicap assessment:				
Low, *n*		*15*		
High, *n*		*19*		
Chronic pain classification:				
I low intensity, *n*		*2*		
II high intensity, *n*		*13*		
III moderate, *n*		*19*		
IV severe, *n*		*0*		
Migraine headache within last 6 months yes, *n*	*13*	*21*	*0.070* ^2^	***0.003* ^2^**
Depression:	0.8 (0.9)	1.42 (0.9)	***0.000* ^1^**	***0.000* ^1^**
Non-specific physical symptoms including pain-related:	0.79 (0.66)	2 (0.83)	***0.000* ^1^**	***0.000* ^1^**
Non-specific physical symptoms excluding pain-related:	0.71 (0.79)	2 (0.93)	***0.000* ^1^**	***0.000* ^1^**

Unless otherwise stated, the data are expressed as medians (IQR interquartile range); ^1^, Mann–Whitney’s U test; ^2^, Chi^2^ Pearson’s test.

**Table 6 jcm-14-07381-t006:** TMD and jaw function limitation in I or II degree of orofacial chronic pain and III degree of orofacial chronic pain in RA patients.

TMD and Jaw Function Limitation	I or II Degree of Orofacial Pain in RA Patients	III Degree of Orofacial Pain in RA Patients	*p Value*
Clicks and crackles, *n*	*8*	*9*	*1.000* ^1^
Friction and grinding, *n*	*7*	*10*	*1.000* ^1^
Grinding or clenching during at night, *n*	*7*	*15*	*0.075* ^1^
Grinding or clenching during a day, *n*	*7*	*12*	*0.488* ^1^
Tinnitus, *n*	*12*	*15*	*1.000* ^1^
Occlusal changes, *n*	*6*	*6*	*0.724* ^1^
Morning jaws pain or stiffness, *n*	*4*	*5*	*1.000* ^1^
**Number of TMD**	3 (2)	4 (3)	*0.631* ^2^
**Number of jaw function limitations:**	4 (2)	5 (4)	*0.450 ^2^*
Chewing, *n*	*14*	*15*	*0.354* ^1^
Drinking, *n*	*1*	*1*	*1.000* ^1^
Exercises, *n*	*4*	*7*	*0.715* ^1^
Eating hard food, *n*	*9*	*16*	*0.237* ^1^
Eating soft food, *n*	*5*	*6*	*1.000* ^1^
Smiling, *n*	*1*	*2*	*1.000* ^1^
Teeth brushing and face washing, *n*	*1*	*9*	***0.020* ^1^**
Yawning, *n*	*11*	*13*	*1.000* ^1^
Swallowing, *n*	*8*	*10*	*1.000* ^1^
Speaking, *n*	*2*	*2*	*1.000* ^1^
Maintaining facial expression, *n*	*2*	*1*	*0.571* ^1^

Unless otherwise stated, the data are expressed as medians (IQR interquartile range); TMD, temporomandibular disorder; *n,* number; ^1^ Fisher’s exact test; ^2^, Mann–Whitney’s U test.

**Table 7 jcm-14-07381-t007:** Comparison of TMJ noises between RA patients and healthy subjects.

	HS *n* = 40	RA patients *n* = 40
**Clicking or crackles during opening/closing, *n* (%)**	*0* (0)	*21* (52.5)
**TMJ sounds during, *n*:**		
**Opening for RS/LS**		
Clear crepitation	*0/0*	*10/7*
Slight crepitation	*0/0*	*18/15*
Cracking	*0/0*	*8/4*
**Closing for RS/LS**		
Clear crepitation	*0/0*	*10/8*
Slight crepitation	*0/0*	*16/7*
Cracking	*0/0*	*7/3*
**Right laterotrusion for RS/LS**		
Clear crepitation	*0/0*	*6/4*
Slight crepitation	*0/0*	*17/16*
Cracking	*0/0*	*3/1*
**Left laterotrusion for RS/LS**		
Clear crepitation	*0/0*	*4/5*
Slight crepitation	*0/0*	*16/15*
Cracking	*0/0*	*3/2*
**Protrusion for RS/LS**		
Clear crepitation	*0/0*	*11/7*
Slight crepitation	*0/0*	*19/25*
Cracking	*0/0*	*5/2*

**Table 8 jcm-14-07381-t008:** Comparison of mandibular kinematics and movement restriction between RA patients and healthy subjects.

	HS *n* = 40	RA Patients *n* = 40	*p* Value	*p Value Adjusted to Age and Gender*
Vertical range of motion [mm]:				
Opening without pain	35 (3.5)	31 (5.5)	***0.000* ^1^**	***0.000* ^1^**
Maximal active opening	37 (3)	33.5 (4.5)	***0.000* ^1^**	***0.000* ^1^**
mean	36 mm	32 mm		
Maximal passive opening	34 (3.5)	34 (5)	*0.437* ^2^	*0.560* ^2^
mean	33.2 mm	33.8 mm		
Vertical inter-incisal distance [mm]	5 (1)	4 (0)	***0.000* ^3^**	***0.000* ^3^**
Horizontal inter-incisal distance [mm]	4 (1)	4 (1)	*0.238* ^3^	*0.132* ^3^
Mandibular midline deviation, *n*:				
Right	*20*	*18*		
Left	*20*	*22*	*0.654* ^4^	*0.791*
Right laterotrusion [mm]:	4 (1)	4 (2)	***0.013* ^3^**	
Left laterotrusion [mm]:	4.5 (1.5)	3 (1)	***0.000* ^3^**	
Protrusion [mm]:	5 (1)	4 (1.5)	***0.000* ^3^**	

Unless otherwise stated, the data are expressed as medians (IQR interquartile range); *n*, number; ^1^, Welch’s test; ^2^, *t*-test; ^3^, Mann–Whitney’s U test; ^4^, Chi^2^ Pearson’s test.

## Data Availability

The original contributions presented in this study are included in the article. Further inquiries can be directed to the corresponding author.

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
