# Peer review of "Temporomandibular Disorders in Patients with Rheumatoid Arthritis"

_jcm, 2025, doi:10.3390/jcm14207381_

Round 1
Reviewer 1 Report
Comments and Suggestions for Authors
I would like to congratulate the authors on their manuscript “temporomandibular disorders in patients with rheumatoid arthritis.”
I read your manuscript with interest and have some comments and suggestions.
Abstract:
- conclusion – I agree TMD screening should be included, but it would be good to mention why – I assume so these patients can receive TMD specific treatment to relieve their TMD? There is no reason why you should include the screening at this moment.
Introduction:
- It would be good to include that the only way to diagnosis active TMJ arthritis in RA would be to perform a contrast – enhanced MRI – which is costly, time consuming and not readily available everywhere.
Materials and Methods:
- where were your HC recruited from? If they were recruited through the same clinic they clearly had some MSK concerns and are not the ideal controls. Can you add a sentence where your HC were recruited from so it is clear if there might have been any bias in this group.
Results:
- Table 4 – I would propose to use short terminology for the different muscles such as Lat Pterygoid with a description at the bottom to make the table easier to read.
- As pain amplification is a known problem in inflammatory arthritis and TMD complaints and pain amplification are hard to distinguish it would have been of interest if there would have been a general pain questionnaire to see if RA patients with TMD had overall more pain sensation than RA patients without TMD as I suspect there will be a relationship. Do you have data on this?
Discussion:
- The relationship with CRP should be discussed in more detail in the discussion – increased CRP is not a finding in all RA patients so a correlation with CRP and TMD issues should be explored more to say if being more inflammatory at itself puts a patient at risk of more TMD?
- The discussion is very long and although very applicable I would advise to try to combine some of the paragraphs.
- In the limitations of your study please include that you do not have any MRI and you can not correlate the TMD findings to real TMJ arthritis and can only speculate on TMJ involvement as the official term for TMJ involvement is that the symptoms/features are due to earlier TMJ arthritis which does not have to be active – in your study you do not know who has real TMJ arthritis in the present or past so you can speculate TMD is associated to TMJ involvement but it could also be associated to pain syndromes and not to sequalae from TMJ arthritis.
Author Response
Thank you very much for your comments and suggestions. In response to your review please find our answers below.
Ad 1. Justification of need to perform TMD screening programme in RA patients has been clarified in the abstract subsection.
Ad 2. A role of enhanced MRI in diagnosing of TMJ has been marked.
Ad 3. One of the author of the study is a physiotherapist and works in Dental Office. All participants who were included in HS group have been recruited from Dental Office. The data regarding the place where HS were recruited have been added.
Ad 4. The suggested abbreviations of lateral and medial pterygoid muscles have been used in the Table 4 and explained at the bottom of the Table 4.
Ad 5. The new possible associations between TMD and overall pain amplification in RA patients have been added and introduced into the study. The detailed data have been presented in Table 6
Ad 6. In the discussion subsection the potential correlation between increased CRP and TMD has been more explored and supported by the new references.
Ad 7. The discussion has been shortened
Ad 8. A lack of MRI in the study has been added as a significant limitation of the study and more. discussed.
Reviewer 2 Report
Comments and Suggestions for Authors
Dear Authors,
first of all, I thank You for giving me the opportunity to read this Your cross-sectional study.
Here are my comments and suggestions. I hope that they useful for You and constructive.
The methodology You followed seems unacceptable to me.
Table 2:
a) there was a gender discrepancy between HS (21/19 F/M) and RA patients (34/6 F/M). Could it be a recruitment bias? An evaluation of the potential importance of gender in the final results was lacking.
b) comorbidities should not have been limited to circulatory and respiratory problems. And then, why these comorbidities and not others?
Table 4:
In the HS group, TMD symptoms were limited almost exclusively to muscle pain during jaw movement. The most painful muscles in RA patients were medial pterygoid muscles, anterior belly of digastric muscle and tendon of temporal muscle. What's the point of comparing such very low numbers?
Discussion:
It seemed too long compared to the scarcity of data.
Author Response
Thank you very much for your comments and suggestions. In response to your review please find our answers below.
Ad 1. An evaluation of the potential importance of gender and age in the final results has been added. The p value adjusted to age and gender has been introduced in the new columns in the tables.
Ad 2. The data related to comorbidities have been completed
Ad 3. The presented results and comparison between muscles are the consequences of the used questionnaire . Examination of these muscles is included into questionnaire.
Ad 4. The discussion has been shortened
Round 2
Reviewer 2 Report
Comments and Suggestions for Authors
Dear Authors,
I read the newer, revised version of Your manuscript.
All my comments and suggestions were satisfactorily met, no doubt. However, the statistical data You entered in the Tables with respect to sex and gender deserved to be discussed and reported in Your conclusions.
Author Response
Thank you very much for your comments and suggestions. In response to your review please find our answers below.
The statistical data related to age and gender have been discussed and supported by the additional relevant references. The new conclusion based on these data has been introduced into the manuscript.
